# Detection of Tick-Borne Pathogens in Ticks from Cattle in Western Highlands of Cameroon

**DOI:** 10.3390/microorganisms10101957

**Published:** 2022-09-30

**Authors:** Yannick Ngnindji-Youdje, Adama Zan Diarra, Michel Lontsi-Demano, Timoléon Tchuinkam, Philippe Parola

**Affiliations:** 1Aix Marseille Univ, IRD, AP-HM, SSA, VITROME, 13005 Marseille, France; 2IHU-Méditerranée Infection, 19-21 Boulevard Jean Moulin, 13005 Marseille, France; 3Vector-Borne Diseases Laboratory of the Applied Biology and Ecology Research Unit (VBID-URBEA), Department of Animal Biology, Faculty of Science of the University of Dschang, Dschang P.O. Box 067, Cameroon

**Keywords:** tick-borne pathogen, zoonoses, cattle, ticks, PCR, Cameroon

## Abstract

This study aimed to detect and identify microorganisms in ticks collected in the Western Highlands of Cameroon. Quantitative real-time and standard PCR assays, coupled with sequencing, were used. A total of 944 ticks collected from cattle in five distinct sites in Cameroon were selected for the analyses. They belonged to five genera (*Amblyomma*, *Hyalomma*, *Rhipicephalus*, *Haemaphysalis,* and *Ixodes*) and twelve species. Real-time PCR revealed that 23% (*n* = 218) of the ticks were positive for *Rickettsia* spp., 15% (*n* = 141) for bacteria of the Anaplasmataceae family, 3% (*n* = 29) for Piroplasmida, 0.5% (*n* = 5) for *Coxiella burnetii,* 0.4% (*n* = 4) for *Borrelia* spp., and 0.2% (*n* = 2) for *Bartonella* spp. The co-infection rate (3.4%, *n* = 32) involved mainly *Rickettsia* spp. and Anaplasmataceae. Of the *Rickettsia* spp. positive ticks, the targeted PCR and sequencing yielded *Rickettsia africae* (78.9%)*, Rickettsia aeschlimannii* (6.4%), *Rickettsia massiliae* (7.8%), *Candidatus* Rickettsia barbariae (0.9%), and *Rickettsia* sp. (0.9%). Anaplasmataceae included *Anaplasma marginale* (4.3%), *Anaplasma platys* (1.4%), *Anaplasma centrale* (0.7%), *Ehrlichia ruminantium* (0.7%), *Wolbachia* sp., *Candidatus* Ehrlichia rustica (13.5%), *Candidatus* Ehrlichia urmitei (7%), and an uncultured *Ehrlichia* sp. (4.2%). *Borrelia theileri* was identified in one *Rhipicephalus microplus* tick. Unfortunately, Piroplasmida could not be identified to the species level. This study demonstrates that in Cameroon, ticks harbour a wide variety of microorganisms and present a risk of zoonotic diseases.

## 1. Introduction

Ticks are obligate haematophagous arachnids, which are distributed globally and parasitise a wide range of vertebrates [1]. On a global scale, ticks are currently considered as the leading vectors of animal diseases and second biggest vectors of human diseases after mosquitoes [2,3]. Only about 20% of the world’s cattle population remains unaffected by ticks and tick-borne diseases. However, the annual global cost losses are estimated to range from USD 14–19 billion [4,5]. With an estimated population of 6.5 million, cattle are regarded in Cameroon as the main source of animal protein in most households, according to the National Institute of the Statistic [6]. However, one of the most important constraints of small- and large-scale cattle production is the high prevalence of infectious diseases [7]. Of the about 900 currently known tick species, only about 10% are of significant veterinary and/or medical importance [8]. In sub-Saharan Africa, ticks of economic importance in livestock belong mainly to four genera, according to the previous studies, namely *Amblyomma*, *Hyalomma*, *Haemaphysalis,* and *Rhipicephalus* [9,10].

The direct effects of tick on their hosts are irritation, anaemia, inflammation, paralysis, abscesses, allergies, hypersensitivity, and skin deterioration at the biting site, which often leads to reduction in weight gain and milk yield [11]. Ticks are also responsible for indirect effects through the transmission of pathogens responsible for bacterial, viral, and protozoan diseases worldwide [12,13,14].

In certain major livestock production zones in Cameroon, particularly the Sudano-Sahelian zone, the high Guinean savannah zone, and the Western Highlands zone, ticks are less studied than other disease vectors. There is also little information on the biodiversity of ticks and microorganisms transmitted, although some studies have been conducted in the country [15,16,17]. There is a lack of support for livestock farmers from veterinarians, which leads to an absence of care and, therefore, the ineffectiveness of control strategies undertaken by livestock farmers. However, in the light of recent information, the prevalence of ticks and tick-borne diseases in livestock is constantly increasing [18]. Furthermore, according to a study conducted in 1982 in the principal cattle-rearing agro-ecological zone of Cameroon (Wakwa research station, Adamawa), approximately 63% of animal mortality was attributed to ticks and tick-borne diseases [19]. The situation was made worse by the recent introduction of the invasive cattle tick (*Rhipicephalus microplus*) into the country whose presence was doubtful prior to 2019 [20,21]. This cattle tick is known as the most significant parasite and disease vector of livestock worldwide. It is known to be the main vector of *Babesia bovis*, *Babesia bigemina*, and *Anaplasma marginale*, causing bovine babesiosis and anaplasmosis in cattle [10,22,23,24]. Currently, the control of tick-borne diseases relies mainly on tick control. One of the difficulties that prevents the eradication of tick-borne diseases is that there is no vaccine, despite enormous efforts employed in chemical vector control. However, as some studies show, very little progress has been made to control ectoparasite infestations in animals using the vaccine approach [25].

This study aimed to detect certain classic tick-associated bacteria in ticks species collected from the Western Highlands of Cameroon using molecular tools.

## 2. Materials and Methods

### 2.1. Study Area

This study was done in the Western Highlands of Cameroon, which are one of the three major agro-ecological zones for livestock production (Figure 1). It is in the mid- and high-latitude zones of the country. The annual average temperature is 20.6 °C and annual precipitation ranges from 1300 to 3000 mm. Two seasons can be distinguished as follows: the rainy season lasts eight months (March to October) and the dry season from November to February [20]. The choice of this zone is justified by the potential for the development of breeding livestock, which are the main host of ticks in Cameroon and Central Africa. In addition, there is a significant livestock trade network between this area and the Adamawa region, which is the primary livestock-producing region in Cameroon. It should also be noted that this area is one of the entry points for animals from West Africa, particularly neighbouring Nigeria [20,26].

### 2.2. Tick Collection and Morphological Identification

Cattle dwellings were visited in five sites in the Western Highlands (zone III), namely Dschang and Nkong-Ni (Bafou) in the Menoua division and Kouoptamo, Massangam, and Koutaba in the Noun division. Ticks were collected from cattle using blunt steel forceps and placed inside a collection tube containing 70% ethanol.

Ticks were first identified morphologically to the species by PhD-trained entomologists from the Vector Borne Disease Laboratory (VBID) at the University of Dschang in Dschang, Cameroon [10,27]. The samples were sent to France after obtaining import authorisation (number FR13-2020 from the French Ministry of Agriculture, Food and Forestry). At the IHU Méditerranée Infection in Marseille (France), the tick morphology was then rechecked using a Leica binocular lens (Leica Camera, Wetzlar, Germany) with an LED light source, by entomologists using previously established taxonomic identification keys [10]. Due to their engorged status and to morphological similarities between the ticks, some specimens in the *Rhipicephalus* genus were identified to the genus level only. The identification of tick specimens was then confirmed by molecular biology and further refined by MALDI-TOF mass spectrometry (Ngnindji et al., under review).

### 2.3. DNA Extraction and Molecular Detection of Microorganisms in Ticks Using Real-Time PCR

DNA was extracted from half of the ticks, as described previously [28]. To investigate the presence of pathogens using primers and probes targeting Anaplasmataceae bacteria, *Bartonella* spp., *Borrelia* spp., Piroplasmida, *Rickettsia* spp., and *Coxiella burnetii*, quantitative PCR (qPCR) was performed on the extracted DNA, using a CFX96 touch detection system (Bio-Rad, Marnes-la-Coquette, France). The reaction mix quantity and the programme for qPCR was the same as previously described [29]. For each qPCR run, DNA from *Rickettsia montanensis*, *Bartonella elizabethae*, *Ehrlichia canis*, *C. burnetii*, *Borrelia crocidurae*, and *Theileria orientalis* were used as a positive control. DNA from *Rhipicephalus sanguineus* s.l. raised in our laboratory, which were free of microorganisms, were used as negative controls. The qPCR tests were considered positive when the cycle threshold (Ct) was lower than 36 [28]. Positive samples for *Rickettsia* spp. were then submitted to the qPCR system, specifically for detecting *Rickettsia africae* [30]. Samples which were positive for *C. burnetii* using IS1111 gene were submitted to the second gene (*IS30A*) for confirmation. In our reference centre for Q fever and Rickettsial infections, only samples positive for both genes will be considered positive for *C. burnetii* [31]. The primers and probes used for quantitative real-time and conventional PCRs are summarised in Table 1.

### 2.4. Standard PCR, Sequencing, and Phylogenetic Analysis

Standard PCR was performed using a thermal cycler (Applied Biosystems, Paris, France) on the samples that were positive in qPCR, and then sequencing to identify the microorganism species. Samples that were negative for *R. africae* but positive for *Rickettsia* spp. were subjected to standard PCR to amplify a 632-bp fragment of the *ompA* gene [34,36]. The amplicons were sequenced, assembled, and compared to GenBank entries sequences by a BLAST search to identify the *Rickettsia* species. For sequences with a low percentage identity with the corresponding sequences in GenBank, PCR targeting a second gene (*gltA*) was used to amplify a 700 bp DNA fragment, followed by sequencing. Samples that were positive for the Anaplasmataceae family were subjected to amplifying and sequencing of 520 bp fragment of the 23S rRNA gene [32]. The samples positive for *Borrelia* spp. under qPCR were subjected to amplifying and sequencing of a 300 bp fragment of the flagellin (*FlaB*) gene [40]. Samples that were Piroplasmida-positive following qPCR were subjected to amplifying and sequencing of a 969 bp fragment of the 18S rRNA gene [29]. The obtained sequences were assembled and analysed using the CromasPro software (version 1.7.7) (Technelysium Pty. Ltd., Tewantin, Australia) and were then blasted against the reference sequences available in GenBank (http://blast.ncbi.nlm.nih.gov/ accessed on 20 July 2021).

Sequences from microorganisms were aligned using the BioEdit v 7.2.5.0 software (University of North Texas, Denton, TX, USA). The aligned sequences were imported into TOPALi v2.5 software (Biomathematics and Statistics Scotland, Edinburgh, UK) and phylogenetic trees were constructed using TOPALi v2.5 software [41]. The maximum likelihood (ML) phylogenetic tree model proposed by default by the software was used to construct the phylogenetic tree. Node numbers are percentages of bootstrap values obtained by repeating 100 interactions of the analysis to generate a majority consensus tree (only those with values equal to greater than 80 were retained).

## 3. Results

### 3.1. Ticks

A total of 944 of the 1483 ticks collected from cattle in five sites of the Western Highlands of Cameroon were randomly selected for the analyses. The combination of the three identification methods allowed us to classify the ticks into 5 genera and 12 species. These included 299 (31.7%) *Rhipicephalus microplus*, 272 (29%) *Rhipicephalus lunulatus*, 217 (23%) *Amblyomma variegatum*, 48 (5%) *Rhipicephalus sanguineus* s.l, 43 (4.5%) *Haemaphysalis leachi* group specimens, 25 (2.6%) *Hyalomma truncatum*, 16 (1.7%) *Hyalomma rufipes*, 12 (1.3%) *Rhipicephalus muhsamae*, 6 (0.6%) *Rhipicephalus annulatus*, 3 (0.3%) *Rhipicephalus decoloratus*, and 3 (0.3%) *Ixodes rasus* (Table 2).

### 3.2. Detection of Microorganisms in Ticks

Of the 944 ticks, 399 (42.6%) were qRT-PCR positive for at least one of the microorganisms tested. Among them were 218 (54.6%) *Rickettsia* spp., 141 (35.3%) Anaplasmataceae, 29 (7.3%) Piroplasmida, 5 (1.2%) *C. burnetii*, 4 (1%) *Borrelia* spp., and 2 (0.5%) *Bartonella* spp. (Table 3).

Among the ticks infected by *Rickettsia* spp., *R. africae* was found in 172 samples (78.9%). *Rickettsia africae* was detected in *Am. variegatum*, *Hy. Truncatum,* and *Rh. microplus*. The remaining 46 tick samples, which were positive for *Rickettsia* spp. but negative for *R. africae*, were subjected to amplification of the *ompA* gene fragment to identify these *Rickettsia* species. Of these 46 positive tick samples, amplification and sequencing provided sequences for 35 (78.2%) samples. The BLAST analyses showed that 17 (48.6%) sequences were 99–100% identical to *Rickettsia massiliae* (MH549236, MN811608), and 14 (40%) sequences were 99–100% identical to the *Rickettsia aeschlimannii* (MH932060, MK922621). Similarly, two (5.7%) sequences were 99.49% identical to *Candidatus* Rickettsia barbariae (KU645284). In contrast, two (5.7%) sequences of *Rickettsia* sp. were 97.88% and 97.92% identical to *Rickettsia slovaca* (MZ851192) using the *ompA* gene. When the citrate synthase (*gltA*) gene was targeted for the same samples, the obtained sequences were 98.09% identical to those of *Rickettsia parkeri* (CP040325). The sequences of the latter species have been deposited in GenBank as *Rickettsia* sp. under the numbers: OP223189, OP223190, OP223191, and OP223192.

*Rickettsia**massiliae* was detected in *Rh. lunulatus* and *Rh. muhsamae*. *Rickettsia aeschlimannii* was observed in *Hy. rufipes*, *Hy. Truncatum*, and *Rh. sanguineus*. *Candidatus* Rickettsia barbariae was detected in *Rh. muhsamae*. Finally, *Rickettsia* sp. was detected in two *Rh. lunulatus* ticks (Table 4).

A total of 141 (35.3%) ticks tested positive by qPCR for bacteria from the Anaplasmataceae family. The amplification of the 23S rRNA gene and sequencing were successful for only 46 (32.6%) samples. The BLAST analysis showed that 19 (41.3%) of the obtained sequences were 98–100% identical to *Candidatus* Ehrlichia rustica (KT364330, MN614109), ten (21.7%) sequences were 99–100% identical to *Candidatus* Ehrlichia urmitei (GenBank KT364334). Six (13%) sequences were 99–100% identical to sequence of *A. marginale* (CP023731), six (13%) sequences were 99–100% identical to an uncultured *Ehrlichia* sp. (MW850476, MK942565), and two (4.3%) sequences were 98.72% and 98.74% identical to *A. platys* (MN626395). Similarly, two sequences were 100% and 99.58% identical to *E. ruminantium* (CR925677) and *A. centrale* (MH321193), respectively. One sequence of *Wolbachia* sp. from *Ha. leachi* was 99.78% identical to *Wolbachia pipientis* (KT827385) for the first hit and other *Wolbachia* from *Mycopsylla fici* deposited in GenBank as “endosymbiont” (KT273261). *Candidatus* Ehrlichia rustica was detected in *Rh. lunulatus*, *Rh. Microplus*, and *Haemaphysalis* sp.; *Candidatus* Ehrlichia urmitei were found in *Rh. microplus.* Uncultured *Ehrlichia* sp. were detected in *Rh. microplus*, *Rh. lunulatus*, *Rh. Sanguinues*, and *Haemaphysalis* sp.; *Anaplasma marginale* were detected in *Rh. microplus* and *Haemaphysalis* sp.; *A. platys* in *Rh. microplus* and *Rh. sanguineus* (Table 4).

*Coxiella burnetii* was detected in five (0.5%) of the samples with *IS1111* and *IS30A* genes. The tick species *Hy. truncatum*, *Hy. Rufipes*, and *R. lunulatus* carried *C. burnetii* DNA.

DNA of *Borrelia* spp. and *Bartonella* spp. were found by qPCR for four and two ticks, respectively. For *Borrelia* spp., we succeeded in amplifying the *flaB* gene sequence only in one of four ticks. A BLAST search showed that sequence was 100% identical with *Borrelia theileri* (MK984606). This *Borrelia* specie was detected in a *Ha. leachi* specimen. However, all the standard PCR to identify the *Bartonella* species failed.

DNA from the Piroplamida was detected in 7.3% (29/399) of positive ticks (17 *Rh. microplus*; 7 *Rh. lunulatus*; 3 *Am. variegatum*; 1 *Hy. rufipes* and 1 *Ha. leachi*.) by qPCR. Unfortunately, we could not amplify these positive samples using standard PCR.

Finally, 29 co-infections (7.3%, 29/399) were detected. Most co-infections (72.4%, 21/29) involved the presence of *Rickettsia* spp. in *Am. variegatum* ticks. Eleven co-infections (38%, 11/29) were observed with *R. africae*, of which nine of the eleven were *R. africae* with *Anaplasma* spp. and two of the eleven were *R. africae* and Piroplasmida. Other co-infections were *Rickettsia* sp. plus *Anaplasma* spp. (17.2%, 5/29) in *Rh. lunulatus*, *Rh. microplus* and *Rh. sanguineus. Rickettsia africae* plus *C. burnetii* (17.2%, 5/29) in *Hy*. *truncatum* and *Hy. rufipes*. *Anaplasma* spp. plus Piroplasmida (17.2%, 5/29) in *Rh. microplus*, *Rh. Lunulatus*, and *Hy. truncatum*.

No bacterial DNA was identified in *I. rasus*, *Rh. Annulatus*, and *Rh. decoloratus* ticks.

Two phylogenetic trees of Rickettsiae and Anaplasmataceae were constructed from the partial sequences of the *ompA* gene and the *23S* rRNA gene sequences of our amplicons, respectively. These phylogenetic trees showed that sequences of the microorganisms detected in this study are close to their homologues available is GenBank, except for two sequences of *Rickettsia* sp. (Figure 2 and Figure 3).

## 4. Discussion

Several species of microorganisms were detected in ticks from Cameroon. DNA of *Rickettsia* spp. was found in 23% of the tested ticks, 78.8% of which was found in *Am. variegatum* and 92.2% was *R. africae*. *Rickettsia africae* is the aetiological agent of African tick-bite fever in humans [42,43]. The results show that the *R. africae* infection rate is high among *Am. variegatum* in the Western Highlands zone of Cameroon. This high prevalence of *R. africae* had already been identified in Cameroon [16,36]. This finding confirms those of previous studies conducted in other African countries [28,44]. *Rickettsia africae* was identified in 7 (6%) of 118 patients with acute fever of unknown aetiology in clinics along the coastal region of Cameroon [16]. *Amblyomma variegatum* is a vector of *R. africae* in sub-Saharan Africa. These ticks are not only vectors but also reservoirs of rickettsiae in sub-Saharan Africa with transstadial and transovarial transmission of *R. africae* infection in *Am. variegatum* ticks [42,45]. *Rickettsia africae* was also detected in 0.7% of *Rh. microplus* and 4.7% of *Hy. truncatum* in this study. These proportions are very low, compared to those obtained in *Am. variegatum.* However, the detection of a microorganism in an arthropod does not mean that this arthropod can act as a vector of the microorganism. This bacterium may have infected *Rh. microplus* and *Hy. truncatum* during co-feeding. Indeed, co-feeding transmission is the transmission, which can occur when ticks (infected and uninfected) feed in close spatial and temporal proximity on the same host. During this form of transmission, the host acts as a belt, bringing together infected and uninfected ticks to facilitate pathogen exchange [46]. Furthermore, it has been shown that in co-feeding transmission (or non-systemic transmission), vector-to-vector transmission on the vertebrate host is essentially immediate [47]. This finding was shown in a previous study in Côte d’Ivoire [44]. In this study, other rickettsial DNA was also found in *Hyalomma* and *Rhipicephalus* tick species. Of the remaining 46 tick samples positives for *Rickettsia* spp., amplification and sequencing yielded sequences for 35 (78.2%) samples. The 11 positive Rickettsia qPCR samples for which amplification with the *ompA* gene was not possible may be due to a higher sensitivity of qPCR, compared to standard PCR [48]. This difference in sensitivity often results in qPCR positive samples with a high Ct (low bacterial load) not being amplified.

*Rickettsia aeschlimannii* was also identified in 75% of *Hy. rufipes* and 4% of *Hy. truncatum*. This microorganism is a recognised human pathogen, causing spotted fever and has been detected in many countries in sub-Saharan Africa, including Cameroon [28,36,42,44]. *Rickettsia aeschlimannii* was also found in 2.2% of *Rh. sanguineus* sl. This tick species is, however, not known to be a competent vector of *R. aeschlimannii*, although it is a rickettsia that has been frequently associated with *Hyalomma* spp. Co-feeding could, therefore, be the cause of infection of this tick species, as mentioned above.

In *Rh. lunulatus* (5.6%) and *Rh. muhsamae* (10%) samples, rickettsial sequences with homology to *R. massiliae* were identified. *Rickettsia massiliae* is a pathogenic rickettsia that is associated with *Rhipicephalus* spp. ticks. It has been described as a human pathogen in Europe and South America, but there has never been reports of human infections in Africa [42]. This is the second finding of *R. massiliae* in Cameroon, following a previous study conducted by Vanegas and collaborators in 2018 [36], and is the first in the Western Highlands area of the country. It was previously reported in *Rh. lunulatus* from Cameroon, in *Rh. senegalensis* (33%) from Côte d’Ivoire and in *Rh. guilhoni* (22%) from Senegal [36,37,44].

In this study, *Candidatus* Rickettsia barbariae, a SFG rickettsiae of unknown pathogenicity, was detected in 20% (2/10) of *Rh. muhsamae*. This *Rickettsia* has previously been reported in Sardinia, Italy, where it was detected in a *Rh. turanicus* ticks and named *Candidatus* Rickettsia barbariae [49]. Several other studies have reported the presence of *Candidatus* Rickettsia barbariae in ticks from livestock in some African countries, including Cameroon [36,50,51].

Two sequences of *Rickettsia* sp. from *Rh. lunulatus* were close to the corresponding sequence of *R. slovaca* (97.92 and 97.88%) using the *ompA* gene, and close to the corresponding sequences of *R. parkeri* at 98.09% similarity with citrate synthase (*gltA*) gene. Furthermore, these two species are not known in sub-Saharan Africa [42]. However, in order to be classified as known species, an isolate should exhibit more than one of the following degrees of nucleotide similarity with the most homologous validated species: 99.9% for the *gltA* gene and, when amplifiable, 98.8% for the *ompA* gene [52]. In this case, it might be a new species of Rickettsia and more work would be needed to characterise it properly.

Several bacteria from the Anaplasmataceae family were detected in our study. Some bacteria from this family are known to be pathogens of human and veterinary importance [53].

We found a *Wolbachia* sp., which was 99.78% identical to *Wolbachia pipientis* (KT827385) and *Wolbachia* endosymbiont at the same percentage identity (KT273261) deposited by Fromont and collaborators in 2015 (unpublished data). *Wolbachia* spp. are known to be obligate intracellular endosymbionts of arthropods [54]. Some studies have shown that environmental factors can influence the presence of *Wolbachia* in arthropods, such as mosquitoes [55]. Furthermore, *Wolbachia* may impact the reproductive biology of their hosts, through a wide range of interactions [56]. The presence of *Wolbachia* sp. has already been reported in several mosquito species in Cameroon [57,58]. This study reports, for the first time in the country, the presence of *Wolbachia* sp. in ticks. The presence of *Wolbachia* spp. in one *Ix. ricinus* has previously been reported in Algeria [59]. However, the mechanism of the transmission of *Wolbachia* in ticks and their consequences on tick biology remains elusive [60,61].

We also identified the bacteria named *Candidatus* Ehrlichia urmitei and *Candidatus* Ehrlichia rustica. The DNA of these two bacterial species were found mainly in *Rh. microplus* and *Rh. lunulatus* ticks in this study. These potential new species have previously been found in *Am. variegatum* and *Rh. microplus* collected from cattle in Côte d’Ivoire [44] and in *Rh. microplus* in Mali [28]. *Candidatus* Ehrlichia urmitei has been also identified in Corsica *R. bursa* ticks [62]. These bacteria have not been characterised to date. Further studies are needed to identify the pathogenicity of these bacteria.

*Anaplasma marginale* and *A. centrale* were detected in *Rh. microplus*. These two are obligate intracellular bacteria responsible for bovine anaplasmosis (gallsickness) worldwide, transmitted by tick species, mainly belonging to the *Rhipicephalus* genus. *Anaplasma marginale* would also be transmitted by *Hy. Rufipes* [10]. The presence of the bacteria *A. marginale* in *Rh. Microplus* is obvious, as this tick species is its principal vector. *Rhipicephalus microplus* is also known to be a competent vector of various tick-borne livestock pathogens, as previously described [20]. It has been shown that the introduction of this invasive tick is most often accompanied by the greatest economic losses in cattle breeding [63]. Several other studies have been conducted in Cameroon showing the presence of these *Anaplasma* spp. Both in ticks and their hosts [15,64,65]. The presence of *A. marginale* has previously been reported in cattle from south-western Ethiopia [66], in *Rhipicephalus* spp. In western Kenya [67], and in Malian *Rh. Microplus* ticks [28].

*Anaplasma platys* was also detected in one *Rh. Microplus* and one *Rh. Sanguineus.* This is a canine anaplasmosis agent, which exclusively infects platelets and causes cyclic thrombocytopenia in dogs. The main known vector is *Rh. sanguineus* s.l. *Anaplasma platys* has also been identified in other mammals, including cattle and humans, and ticks worldwide [68]. This *Anaplasma* has already been detected from blood samples from cattle in Cameroon [17], but our study marks the first time it has been discovered in *Rh. sanguineus* s.l. ticks from Cameroon. On the scale of Africa as a whole, *A*. *platys* is a species frequently found in the northern part of the continent. However, it had been previously detected in cattle from Nigeria [69]. The presence of *A*. *platys* in main vector in this area poses a risk to potential host dogs.

*Ehrlichia ruminantium* is the aetiological agent of heartwater or cowdriosis, which particularly affects domestic and wild ruminants. It is transmitted by the ticks of the genus *Amblyomma*, primarily *Am. variegatum* and *Am. hebraeum* [70]. The prevalence of 0.7% in main vector, *Am. variegatum*, was significantly lower in comparison to recently published data (8.3%) from Mali [28]. *Ehrlichia ruminantium* had already been reported in several studies conducted in Cameroon, both in ticks and on their hosts [16,17,71,72]. These bacteria were previously described in *Am. variegatum* from several countries in western Africa [28,44].

We found *Borrelia theileri*, a member of the tick-borne relapsing fever group [45]. This study reports, for the first time, the presence of *B*. *theileri* in ticks from Cameroon. This had already been detected in cattle blood in the country [17]. *Rhipicephalus microplus*, in which this bacterium was found in this study, is genetically related and overlaps in distribution with *Rh. decoloratus* and *Rh. annulatus*, the main known vectors of *B. theileri* [73]. The infection rate (0.3%) reported here is comparable to the study in Mali, which found 0.5% infected with *B. theileri* in *Rh. geigyi* collected from cattle. Nevertheless, the vector capacity of *Rh. geigyi* is unknown [74]. Other studies have revealed the presence of species of this genus in different tick species in Algeria [59], Mali [28], and Côte d’Ivoire [44]. Reported cases of tick-borne relapsing fever have proven to be responsible for economic losses in livestock [75].

For some of our tick samples, which were positive for *Bartonella* spp. by qPCR using the *ITS2* genes and Piroplasmida using *18S Piro* gene, no amplification was observed after standard PCR, despite the fact that our DNA quantification gave satisfactory results. This issue may be due to a higher sensitivity of qPCR, compared to conventional PCR [48]. It could also be due to DNA degradation or to PCR inhibition by cattle blood. However, several other microorganisms in the Piroplasmida order have already been found in cattle blood samples from Cameroon [17,64,76].

*Coxiella burnetii*, a strict intracellular Gram-negative bacterium, is responsible for Q fever affecting humans and a variety of animals [77]. Although Q fever is far more frequently airborne, at least seven hard and soft tick species, including *Hyalomma* spp., have formally been shown to be competent vectors of *C. burnetii* [78]. Our study is the first to detect *C. burnetii* in tick samples (0.5%) in Cameroon. The prevalence rate is comparable to the 0.6% rate reported in ticks from Côte d’Ivoire [44]. In addition, a study conducted in Mali and Nigeria reported a high prevalence of 37.6% and 14% in ticks, respectively [28,79]. *Coxiella burnetii* represents a real, albeit underappreciated threat to human and animal health throughout Africa [77]

Finally, it has been reported that ticks are often co-infected after taking a blood meal from a host carrying several infectious agents [80]. Several studies have reported mixed infections in feeding ticks caused mainly by *Rickettsia* spp. and *C. burnetii* [44,79]. We also reported, for the first time, mixed infections in ticks from Cameroon involving mainly *R. africae* and *Ehrlichia* spp. The co-infected rate here (3.4%) is comparable to that (1.3%) observed in Nigeria [79].

## 5. Conclusions

This study demonstrates that ticks in Cameroon harbour a wide variety of tick-borne pathogens of veterinary and medical importance, in particular *R. africae*, the agent of ATBF. Humans are thus at risk of infection with *R. africae*, and Africa tick-bite fever should also be considered in patients presenting with febrile illnesses. This study also reports, for the first time in ticks from Cameroon, the presence of various agents, such as *Borrelia theileri*. Nevertheless, further studies are needed to ascertain the origin and zoonotic potential of the strains and their significance for animals and human health. These data can contribute towards future research, for example by providing an avenue for larger studies of ticks and the pathogens they harbour in Cameroon.

## Figures and Tables

**Figure 1 microorganisms-10-01957-f001:**
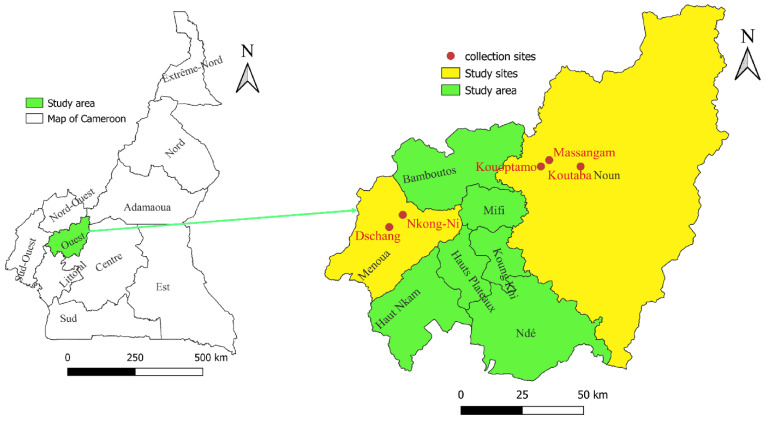
Map of the Menoua and Noun Divisions in the Western Region of Cameroon showing tick collection sites from cattle.

**Figure 2 microorganisms-10-01957-f002:**
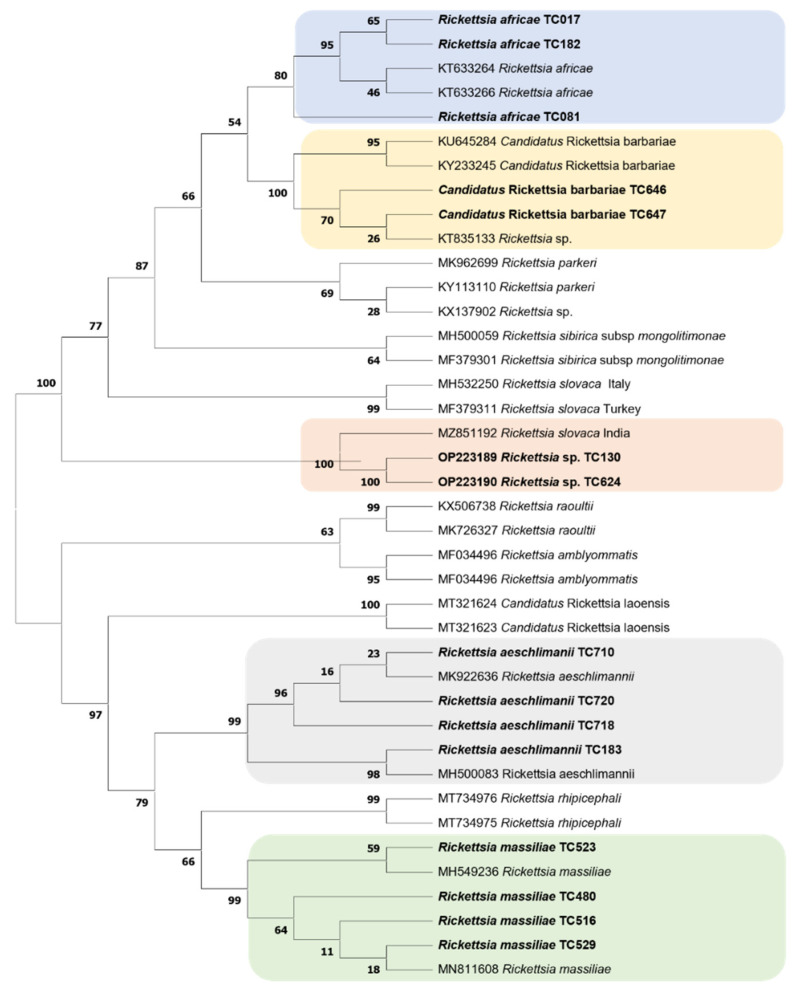
Phylogenetic tree of *Rickettsia* species detected in ticks from the Western Highlands of Cameroon and constructed by maximum likelihood method. Partial sequence of *ompA* gene from ticks collected in 2018 were aligned. Microorganisms sequenced in the current study are in bold.

**Figure 3 microorganisms-10-01957-f003:**
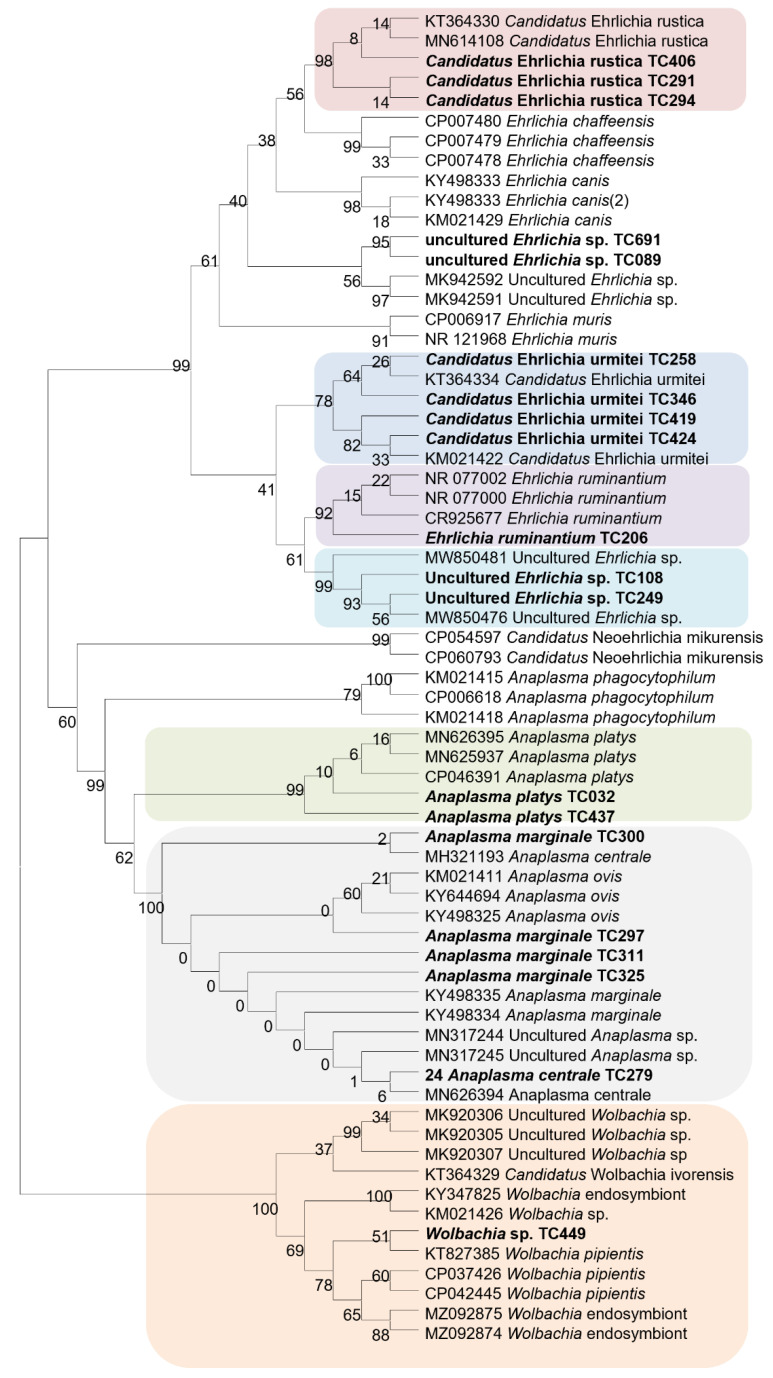
Phylogenetic tree of Anaplasmataceae detected in ticks from the Western Highlands of Cameroon Maximum likelihood method. The partial sequence of *23S* rRNA gene from ticks collected in 2018 were aligned. Microorganisms sequenced in the current study are in bold.

**Table 1 microorganisms-10-01957-t001:** Primers and probes used for quantitative real-time and standard PCR in this study.

Microorganisms	Targeted Sequence	Primers (5′-3′) and Probes (Used for qPCR Screening or Sequencing)	References
Anaplasmataceae	*23S (TtAna)*	f_TGACAGCGTACCTTTTGCAT r_GTAACAGGTTCGGTCCTCCA p_6FAM-GGATTAGACCCGAAACCAAG	[32,33]
	*23S (520-bp)*	f_ATAAGCTGCGGGGAATTGTC r_TGCAAAAGGTACGCTGTCAC	
Piroplasmida	*5.8S*	f_AYYKTYAGCGRTGGATGTC r_TCGCAGRAGTCTKCAAGTC p_FAM-TTYGCTGCGTCCTTCATCGTTGT-MGB	[32]
	*18S (969-bp)*	f1_GCGAATGGCTCATTAIAACA f4_CACATCTAAGGAAGGCAGCA f3_GTAGGGTATTGGCCTACCG * r4_AGGACTACGACGGTATCTGA *	
*Rickettsia*	*gltA (RKND03)*	f_GTGAATGAAAGATTACACTATTTAT r_GTATCTTAGCAATCATTCTAATAGC p_6FAM-CTATTATGCTTGCGGCTGTCGGTTC	[34,35]
	*ITS (Rafricae)*	f_TGCAACACGAAGCACAAAAC r_CCTCTTGCGAAACTCTACTTTTGA 6FAM-CGTGTGGATTCGAGCACCGGA	[30]
	*OmpA (630-bp)*	70_ATGGCGAATATTTCTCCAAAA 701_GTTCCGTTAATGGCAGCATCT 180_GCAGCGATAATGCTGAGTA *	[12,36]
	*gltA (400-bp)*	f_ATGACCAATGAAAATAATAAT r_CTTATACTCTCTATGTACA	
*Borrelia*	*ITS4*	f_GGCTTCGGGTCTACCACATCTA r_CCGGGAGGGGAGTGAAATAG p_6FAM-TGCAAAAGGCACGCCATCACC	[37]
	*flaB (344-bp)*	f_TGGTATGGGAGTTTCTGG r_TAAGCTGACTAATACTAATTACCC	
*Bartonella*	*ITS2*	f_GATGCCGGGGAAGGTTTTC r_GCCTGGGAGGACTTGAACCT p_GCGCGCGCTTGATAAGCGTG	[38]
*Correlia burnetii*	*IS30A*	f_CGCTGACCTACAGAAATATGTCC r_GGGGTAAGTAAATAATACCTTCTGG p_CATGAAGCGATTTATCAATACGTGTATG	[39]
	*IS1111A*	f_CAAGAAACGTATCGCTGTGGC r_CACAGAGCCACCGTATGAATC 6FAM-CCGAGTTCGAAACAATGAGGGCTG	[31]

Abbreviation *, used for sequencing only.

**Table 2 microorganisms-10-01957-t002:** Tick species and numbers of tick specimens collected from cattle in the five study sites in Western Highlands of Cameroon. For each site, the number of female tick specimens is indicated in parentheses.

Tick Genus	Tick Species	No of Ticks Collected	Sex		Menoua Division		Noun Division		
			Male	Female	Nkong-Ni	Dschang	Kouoptamo	Massangam	Koutaba
*Amblyomma*	*Am. variegatum*	353	245	108	80 (30)	87 (42)	35 (6)	67 (16)	84 (14)
*Rhipicephalus*	*Rh. microplus*	552	168	384	207 (153)	101 (62)	104 (83)	88 (56)	52 (30)
	*Rh. annulatus*	6	6	0	0	6	0	0	0
	*Rh. decoloratus*	3	3	0	3	0	0	0	0
	*Rh. lunulatus*	387	239	148	70 (20)	105 (50)	45 (15)	92 (41)	75 (22)
	*Rh. sanguineus*	48	40	8	38 (8)	10 (0)	0	0	0
	*Rh. muhsamae*	10	10	0	7	3	0	0	0
	*Rhipicephalus* spp.	35	0	35	5 (5)	26 (26)	0	4 (4)	0
*Haemaphysalis*	*Ha. leachi*	45	35	10	35 (10)	10 (0)	0	0	0
*Hyalomma*	*Hy. rufipes*	16	10	6	16 (6)	0	0	0	0
	*Hy. truncatum*	25	14	11	25 (11)	0	0	0	0
*Ixodes*	*Ix. rasus*	3	0	3	3		0	0	0
Total		1483	770	713	489	348	184	251	211

**Table 3 microorganisms-10-01957-t003:** Percentage of positive tick species by quantitative real-time PCR.

Tick Species										
Microorganism	Target Sequence	*Am. variegatum*	*Rh. microplus*	*Rh. sanguineus*	*Ha. leachi*	*Rh. lunulatus*	*Rh. muhsamae*	*Hy. rufipes*	*Hy. truncatum*	(%) Pos/Total
*Rickettsia* spp.	*gltA (RKND03)*	78.8% (171/217)	0.6% (2/308)	4.3% (2/46)	-	8.8% (24/272)	35% (3/12)	39% (16/41)		23% (218/944)
*R. africae*	poT15-dam2	77.4% (168/217)	0.6% (2/308)	-	-	-	-	-	8% (2/25)	18.2% (172/944)
Anaplasmatacae	*23SrRNA(TtAna)*	7.4% (16/217)	25% (77/308)	8.3% (4/48)	14% (6/43)	14% (38/272)	-	-	-	14.9% (141/944)
Piroplasmida	*5.8S/Piro 18S*	1.4% (3/217)	5.5% (17/308)	-	2.3% (1/43)	2.6% (7/272)	-	6.3% (1/16)	-	3% (29/944)
*Bartonela* spp.	*(Barto ITS2)/gltA*	-	0.6% (2/308)	-	-	-	-	-	-	0.2% (2/944)
*Borrelia* spp.	*(Bor ITS4)*	0.5% (1/217)	0.6% (2/308)	-	2.3% (1/43)	-	-	-	-	0.4% (4/944)
*C. burnetii*	*(IS1111)/ITS30A*	-	-	-	-	0.3% (1/272)	-	6.3% (1/16)	12% (3/25)	0.5% (5/944)

**Table 4 microorganisms-10-01957-t004:** Tick species collected in Cameroon and studied for microorganisms.

Tick Species											
Microorganism	Target Sequence	Per. Ident (%)	*Am. variegatum*	*Rh. microplus*	*Rh. sanguineus*	*Ha. leachi*	*Rh. lunulatus*	*Rh. muhsamae*	*Hy. rufipes*	*Hy. truncatum*	(%) Sequences Obtained/Pos qPCR
*Rickettsia aeschlimannii*	*ompA*	99.49–100	-	-	50% (1/2)	-	-	-	75% (12/16)	4% (1/25)	6.4% (14/218)
*Rickettsia massiliae*		99.83–100	-	-	-	-	59.2% (16/27)	10% (1/10)		-	7.8% (17/218)
*Candidatus* Rickettsia barbariae		99.49	-	-	-	-	-	20% (2/10)		-	0.9% (2/218)
*Rickettsia* sp.		97.88–97.92	-	-	-	-	7.4% (2/27)	-		-	0.9% (2/218)
*Anaplasma centrale*	*23S Ana*	100	-	1.3% (1/77)	-		-	-	-	-	0.7% (1/141)
*Ehrlichia ruminantium*		100	6.2% (1/16)	-	-		-	-	-	-	0.7% (1/141)
uncultured *Ehrlichia* sp.		98.32–100	-	3.9% (3/77)	25% (1/4)	16.7% (1/6)	2.6% (1/38)	-	-	-	4.2% (6/141)
*Candidatus* Ehrlichia urmitei		99.16–100	-	13% (10/77)	-		-	-	-	-	7% (10/141)
*Anaplasma marginale*		99.79–100	-	6.5% (5/77)	-	16.7% (1/6)	-		-	-	4.3% (6/141)
*Candidatus* Ehrlichia rustica		98.6–100	-	9% (7/77)	-	16.7% (1/6)	28.9% (11/38)	-	-	-	13.5% (19/141)
*Anaplasma platys*		98.72–98.74	-	1.3% (1/77)	25% (1/4)	-	-	-	-	-	1.4% (2/141)
*Wolbachia pipientis*		99.78	-	-	-	16.7% (1/6)	-	-	-	-	0.7% (1/141)
*Borrelia theileri*	*flaB*	100		1.3% (1/77)							25% (1/4)

## Data Availability

Not applicable.

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
