# Peer review of "Detection of Tick-Borne Pathogens in Ticks from Cattle in Western Highlands of Cameroon"

_microorganisms, 2022, doi:10.3390/microorganisms10101957_

Round 1

Reviewer 1 Report

1. The tick speicies name should be labeled with two-letter genus abbreviation because single letter applies to more than one genus in this study.

2. "," should be added after "Wolbachia sp." in abstract.

3. "Boophilus" and the abbreviation "B" should be deleted.

4. "Thee direct effects on their hosts are irritation, anaemia, inflammation,", the first word may be "The".

5. The primers for detection of Rickettsia targeting the gltA gene should be shown. In addition, "Of these 46 positive tick samples, amplification and sequencing provided sequences for 35 (78.2%) samples.", the PCR method should be  applied to other 11 positive tick samples.

6. "from Ha. Leachi", "L" must be a lowercase  "l".

7. "Coxiella burnetii was detected in 25 (6.2%) of positive samples", while "in a second step for confirmation with the IS30A gene only five (1.2%) were positive", the positive rates tested by these two tests are too different. Another method may be needed.

8. "Unfortunately, we could not amplify these positive samples using standard PCR.", other PCR method is needed.

9. "trans-stadial" ? should be "transstadial".

10. "Candidatus, Ehrlichia urmitei", delete the ",".

11. "would also be transmit by", should be "transmitted".

12. many spell mistakes in the text, please correct.

Author Response

Manuscript ID: microorganisms-1888892

Type of manuscript: Article
Title: Detection of tick-borne pathogens in ticks from Cattle in Western
Highlands of Cameroon

Comments and Suggestions for Authors

  1. The tick speicies name should be labeled with two-letter genus abbreviation because single letter applies to more than one genus in this study.

Author’s answer: Thank you the reviewer for this remark, these changes are now well respected throughout the text.

  1. "," should be added after "Wolbachia sp." in abstract.

Author’s answer: The reviewer is right have added the comma after Wolbachia sp.

  1. "Boophilus" and the abbreviation "B" should be deleted.

Author’s answer: We thank the reviewer, “Boophilus” and the abbreviation “B” have been deleted throughout the text.

  1. “Thee direct effects on their hosts are irritation, anaemia, inflammation,”, the first word may be “The”.

Author’s answer: Thank you, this change has been made. The sentence now reads as follows:

« The direct effets of ticks on their hosts are irritation… »

  1. The primers for detection of Rickettsia targeting the gltA gene should be shown. In addition, "Of these 46 positive tick samples, amplification and sequencing provided sequences for 35 (78.2%) samples.", the PCR method should be applied to other 11 positive tick samples.

Author’s answer: We thank the reviewer, the primers for the detection of Rickettsia targeting the gltA gene are now listed in Table 1.

We have performed standard PCR on these 11 samples using ompA gene. However, no amplification was obtained for these samples. We believe that this may be due to a difference in sensitivity between qPCR and standard PCR and we have added a reference. A sentence will be added in discussion section to clarify this follows:

« The 11 positive Rickettsia qPCR samples for which amplification with the ompA gene was not possible may be due to a higher sensitivity of qPCR compared to standard PCR (Kidd et al. 2008). This difference in sensitivity often results in qPCR positive samples with a high Ct (low bacterial load) not being amplified»

  1. "from Ha. Leachi", "L" must be a lowercase "l".

Author’s answer: Thanks for this remark, this error has been corrected: « … from Ha. leachi … »

  1. "Coxiella burnetii was detected in 25 (6.2%) of positive samples", while "in a second step for confirmation with the IS30A gene only five (1.2%) were positive", the positive rates tested by these two tests are too different. Another method may be needed.

Author’s answer: We thank the reviewer, for the detection of Coxiella burnetii, two genes are used (IS1111 and IS30A) and only samples positive for these two genes will be considered positive. The IS1111 gene is the first line gene and therefore more sensitive than the IS30A gene which is the confirmation gene.

To allow a good understanding, some precisions will be brought in the manuscript.

In the materials and methods section:

« In our reference center for Q fever Rickettsial infections, only samples positive for both genes will considered positive for C. burnetii, according to the previous study (Mediannikov et al., 2010) »

In results section

« Coxiella burnetii DNA was detected in 5 (0.5%) of the positive samples by IS1111 and IS30A genes. The tick species Hy. truncatum, Hy. rufipes and R. lunulatus carried C. burnetii DNA »

  1. "Unfortunately, we could not amplify these positive samples using standard PCR.", other PCR method is needed.

Author’s answer: We used the 18S gene for amplification and obtained no bands on migration. Since the 18S gene is the only system that is more discriminating for piroplasmida.

Other reasons for this failure are explained in the discussion section as follows:

«… no amplification was observed after standard PCR, despite the fact that our DNA quantification gave satisfactory results. This issue may be due to a higher sensitivity of qPCR compared to conventional PCR [69]. It could also be due to PCR inhibition by cattle blood. However, several other microorganisms in the piroplasmida order have already been found in cattle blood samples from Cameroon [16,57,70].»

  1. "trans-stadial" ? should be "transstadial".

Author’s answer: Thank: you"trans-stadial" is replaced by "transstadial".

  1. "Candidatus, Ehrlichia urmitei", delete the ",".

Author’s answer: the "comma" was deleted

  1. "would also be transmit by", should be "transmitted".

Author’s answer: Thank you, this error has been corrected

  1. many spell mistakes in the text, please correct.

Author’s answer: We would like to thank you for your very high level of participation. Your comments, suggestions and recommendations have been of great importance in improving the quality of this work.

Furthermore all recommendations, comments, suggestions and remarks mentioned in the pdf text have been taken into account in the new word version of the manuscript

Refference

Kidd L, Maggi R, Diniz PPVP, Hegarty B, Tucker M, Breitschwerdt E. Evaluation of conventional and real-time PCR assays for detection and differentiation of Spotted Fever Group Rickettsia in dog blood. Vet Microbiol. 2008; 129(3–4): 294–303.

Mediannikov, O.; Fenollar, F.; Socolovschi, C.; Diatta, G.; Bassene, H.; Molez, J.-F.; Sokhna, C.; Trape, J.-F.; Raoult, D. Coxiella Burnetii in Humans and Ticks in Rural Senegal. PLoS Negl Trop Dis 2010, 4, e654

Reviewer 2 Report

The study of Ngnindji-Youdje et al. provides new and interesting results on the occurrence and prevalence of pathogenic microorganisms in ticks collected from cattle in Cameroon. The authors explored the collected materials sufficiently and described the obtained results in detail. Thus, the results should be published. However, before considering for acceptance, major revisions are required.

The main shortcoming of the manuscript is the language, which should be profoundly revised by a native speaker, preferably expert in the field. The present manuscript contains a lot of formal errors, errors in grammar and style, is too wordy and repetitive in some places, especially in sections results and discussion. Thus please avoid repetitive parts and shorten the text.

Another shortcoming is the format of tables and low quality of figures. The list of references should be arranged according to instructions for authors.

Some of the errors (but not all) as well as specific comments are included in the attached file.

Author Response

Please see manuscript for requested changes

Round 2

Reviewer 1 Report

This manuscript can be accepted.

Author Response

Manuscript ID: microorganisms-1888892

Type of manuscript: Article
Title: Detection of tick-borne pathogens in ticks from Cattle in Western
Highlands of Cameroon

  • please makie sure that all text is of sufficient size and resolution to be readable

Author’s answer: Thank you the reviewer for this remark, this has been done. And we have corrected the figure by increasing the resolution and the size of the characters

  • I apologize for the error in the previous comments, please correct as indicated

Author’s answer: Thank you, this has been corrected as indicated

  • Please correct the figure caption which should not be part of the figure, but placed below it

Author’s answer: Thanks for this remark, this error has been corrected

  • please check again Latin names and correct where necessary

Author’s answer: Thank you the reviewer for this remark, these changes are now well respected throughout the text.

  • italicise, "phagocytophila" does not start with a capital letter

Author’s answer: Thanks for this remark, this error has been corrected:

  • thuis reference is not numbered, please correct

Author’s answer: thank you, this reference is now numbered

We would like to thank you once against for your very high level of participation. Your comments, suggestions and recommendations have been of great importance in improving the quality of this work.

Reviewer 2 Report

The manuscript has been improved, but still a few corrections are needed - see comments in the attached file

Author Response

(The authors gave the same response as above.)
